# Resveratrol Affects Sphingolipid Metabolism in A549 Lung Adenocarcinoma Cells

**DOI:** 10.3390/ijms231810870

**Published:** 2022-09-17

**Authors:** Albena Momchilova, Roumen Pankov, Galya Staneva, Stefan Pankov, Plamen Krastev, Evgenia Vassileva, Rusina Hazarosova, Nikolai Krastev, Bozhil Robev, Biliana Nikolova, Adriana Pinkas

**Affiliations:** 1Institute of Biophysics and Biomedical Engineering, Bulgarian Academy of Sciences, Acad. G. Bonchev Str. bl.21, 1113 Sofia, Bulgaria; 2Biological Faculty, Sofia University “St. Kliment Ohridki”, 8, Dragan Tzankov Str., 1164 Sofia, Bulgaria; 3Cardiology Clinic, University Hospital “St. Ekaterina”, 1431 Sofia, Bulgaria; 4Clinic of Neurology, Tsaritsa Yoanna University Hospital-ISUL, 1527 Sofia, Bulgaria; 5Department of Anatomy, Histology and Embryology, Medical University—Sofia, Blvd. Sv. Georgi Sofiisky 1, 1431 Sofia, Bulgaria; 6Medical Center Relax, 8 Ami Bue Str., 1606 Sofia, Bulgaria; 7Department of Medical Oncology, University Multi-Profile Hospital for Active Treatment (UMHAT) “St. Ivan Rilski”, 1606 Sofia, Bulgaria; 8CSTEP, Office of Continuing Education, Suffolk County Community College 30 Greene Ave., Sayville, NY 11782, USA

**Keywords:** resveratrol, sphingolipid metabolism, ceramide, sphingosine-1-phosphate, lung cancer cells

## Abstract

Resveratrol is a naturally occurring polyphenol which has various beneficial effects, such as anti-inflammatory, anti-tumor, anti-aging, antioxidant, and neuroprotective effects, among others. The anti-cancer activity of resveratrol has been related to alterations in sphingolipid metabolism. We analyzed the effect of resveratrol on the enzymes responsible for accumulation of the two sphingolipids with highest functional activity—apoptosis promoting ceramide (CER) and proliferation-stimulating sphingosine-1-phosphate (S1P)—in human lung adenocarcinoma A549 cells. Resveratrol treatment induced an increase in CER and sphingosine (SPH) and a decrease in sphingomyelin (SM) and S1P. Our results showed that the most common mode of CER accumulation, through sphingomyelinase-induced hydrolysis of SM, was not responsible for a CER increase despite the reduction in SM in A549 plasma membranes. However, both the activity and the expression of CER synthase 6 were upregulated in resveratrol-treated cells, implying that CER was accumulated as a result of stimulated de novo synthesis. Furthermore, the enzyme responsible for CER hydrolysis, alkaline ceramidase, was not altered, suggesting that it was not related to changes in the CER level. The enzyme maintaining the balance between apoptosis and proliferation, sphingosine kinase 1 (SK1), was downregulated, and its expression was reduced, resulting in a decrease in S1P levels in resveratrol-treated lung adenocarcinoma cells. In addition, incubation of resveratrol-treated A549 cells with the SK1 inhibitors DMS and fingolimod additionally downregulated SK1 without affecting its expression. The present studies provide information concerning the biochemical processes underlying the influence of resveratrol on sphingolipid metabolism in A549 lung cancer cells and reveal possibilities for combined use of polyphenols with specific anti-proliferative agents that could serve as the basis for the development of complex therapeutic strategies.

## 1. Introduction

Lung cancer is one of the most lethal and widely spread types of cancer [1]. It is the leading cause of cancer death in both men and women around the world, and more people are reported to have died from lung cancer compared to any other type of cancer. Notwithstanding the great achievements in anti-tumor therapeutic schemes and approaches, the prognosis for lung cancer patients still remains unfavorable [2]. Thus, the development of new and more effective therapeutic strategies, especially ones involving natural anti-tumor substances, is extremely necessary.

Resveratrol (3,5,4′-trihydroxy-trans-stilbene) is a naturally occurring polyphenol synthesized in various plants, especially in red grapes, peanuts, and some berries, and is also found in red wine and identified as a chemo-preventive agent [3]. It belongs to a class of defense molecules called phytoalexins and is toxic to many plant pathogens [4,5]. A vast range of beneficial effects have been ascribed to resveratrol, such as anti-inflammatory, antioxidant, anti-cancer, and anti-aging effects, among others [6].

There is evidence suggesting that some of the pharmacological properties of resveratrol are mediated through alterations in sphingolipid metabolism. Sphingolipids are involved in many cellular processes, including cell proliferation, signaling, apoptosis, pro- or anti-proliferative pathways, etc. [7]. Some of the most functionally active members of the sphingolipid (SL) family are sphingomyelin (SM), ceramide (CER), sphingosine (SPH), and sphingosine-1-phosphate (S1P) [8]. They have emerged as mediators of cell death and proliferation in cancer and as potential chemotherapeutics [9].

CER is a biologically active sphingolipid that mediates anti-proliferative processes such as apoptosis, cell growth inhibition, senescence modulation, differentiation, and autophagy [9]. CER levels in cells are regulated via de novo synthesis, hydrolysis through the sphingomyelinase pathway, and degradation performed by ceramidases [10,11]. CER has been shown to exhibit significant anti-tumor activity in various cancer types, such as lung, breast, and prostate cancers [12,13,14]. Inhibiting the activity of ceramidases and/or stimulating sphingomyelinases leads to an increase in CER levels, which can effectively prevent resistance to chemotherapeutic drugs and induce apoptosis [15]. Enzymes participating in the SM /S1P pathway and their products play important roles in cancer development and progression [16,17]. CER especially induces cell-type-specific apoptosis by activating protein kinase C, protein phosphatases, and proteases and also modulates the pro-apoptotic Bcl-2-family proteins [18]. On the contrary, S1P acts as an anti-apoptotic agent by stimulating G-protein-coupled receptors activating RAS, RAC, protein kinase B, and phospholipase C (PLC) [9]. In addition, S1P stimulates inflammation by upregulation of cyclooxygenase 2 with overproduction of prostaglandin E2. This sphingolipid participates in the progression of various pathological processes and diseases, including, besides inflammation, oxidative stress, neurodegenerative pathologies, etc., but, most importantly, it supports cell proliferation and survival [19,20]. Thus, sphingolipid metabolism plays an important role in the balance between apoptosis and proliferation in various tumor cells, including lung cancer in general and A549 lung adenocarcinoma cells in particular [21,22,23].

Resveratrol has been reported to alter sphingolipid levels and metabolism through multiple mechanisms. First of all, it increases ceramide to induce apoptosis in colon, breast, and prostate cancer cells. The effect of resveratrol in some cancers is due to increased de novo ceramide biosynthesis, whereas, in others, it stimulates the SM degrading processes [24,25,26]. The mechanisms through which resveratrol induces apoptosis in cancer cells also involves a COX2-dependent pathway or influences sphingosine kinase 1 (SK1) and its product S1P [27,28]. There is also evidence that resveratrol treatment can lead to autophagy by altering sphingolipid metabolism in gastric cancer cells. Thus, resveratrol-induced modulation of ceramide and S1P levels seems to be related to inhibition of cell proliferation and survival and induction of apoptosis through various pathways [29].

Since resveratrol modulates cellular sphingolipids and there are not many studies on its effect in lung adenocarcinoma cells, we carried out investigations on the effect of this polyphenol on sphingolipid metabolism in A549 cells. We analyzed the biochemical mechanisms underlying the influence of resveratrol on the enzymes responsible for the maintenance of the levels of sphingolipids with high physiological activity which determine the balance between pro-apoptotic and pro-survival processes in human lung adenocarcinoma cells.

## 2. Results

### 2.1. Effect of Resveratrol on Sphingolipid Level and Neutral Sphingomyelinase

The influence of resveratrol on SM levels in plasma membranes isolated from lung adenocarcinoma A549 cells is presented in Figure 1. Incubations with resveratrol were performed for 24 h at a concentration of 100 µM. The percentage of SM in the total phospholipids was reduced by 28% as a result of resveratrol treatment.

Since one major reason for SM reduction in membranes could be activation of neutral sphingomyelinase (nSMase), which hydrolyzes SM to CER and phosphocholine, we analyzed this activity in control and resveratrol-treated A549 cells (Figure 2A). We observed a tendency for elevation in nSMase activity as a result of incubation with resveratrol, but the differences between the obtained values were not statistically significant. Additionally, Western blot analysis showed that the expression of nSMase in A549 cells was not affected by resveratrol treatment (Figure 2B).

Since SM levels were reduced in resveratrol-treated cells but nSMase was not activated and did not show increased expression, we analyzed the alterations in SM content in the incubation medium (Figure 3). SM levels were elevated in the medium of resveratrol-treated cells, implying that this polyphenol affected the efflux of SM from the membranes into the surrounding medium. Based on the high affinity between SM and cholesterol (CH) in plasma membranes, where they form signaling platforms called rafts, we also analyzed the changes in CH in the medium before and after resveratrol treatment (Figure 3). Interestingly, CH content was also increased after incubation with resveratrol, suggesting that resveratrol treatment of these cancer cells probably influenced the integrity of the raft domains, because SM and CH are major components of these membrane structures.

Besides SM, other physiologically active sphingolipid metabolites such as ceramide (CER), sphingosine (SPH), and sphingosine-1-phosphate (S1P) were also altered in A549 cells due to resveratrol treatment (Figure 4).

### 2.2. Effect of Resveratrol on the Enzymes Maintaining Ceramide Level

CER is a product of SMase-induced hydrolysis of SM and is one of the most biologically active sphingolipids. This sphingolipid is closely related to apoptosis initiation, which makes it an important member of the sphingolipid family, especially in cancer cells. CER levels were increased (42%) in resveratrol-treated lung adenocarcinoma cells, this being a significant finding with respect to the fate of the treated cancer cells. Another member of the sphingolipid family is SPH, which has less physiological importance compared to CER, but it functions as a precursor of the other very active sphingolipid representative, S1P, the latter being responsible for cell proliferation. The level of SPH was elevated by 25% due to resveratrol treatment, whereas S1P was reduced by 36%. This is an interesting finding, because the balance between these two lipids is maintained by the enzyme sphingosine kinase, which modulates the equilibrium between apoptosis and proliferation.

The level of cellular CER is maintained by three different enzyme activities which are responsible for accumulation and/or elimination of CER molecules. First, as mentioned above, the SMase pathway is a basic cellular source of CER through the degradation of one major phospholipid membrane component, SM. As shown in Figure 2, nSMase activation was very weak, implying that it was not responsible for the observed elevation in CER (42%). Another optional source of CER is through its anabolic pathway, where CER synthase acts as a key enzyme maintaining CER accumulation. Measuring the CER synthase activity, we found that it was upregulated by 38% as a result of resveratrol treatment. (Figure 5A). Western blot analysis showed that the expression of SM synthase was elevated by about 35% due to resveratrol action (Figure 5B). In these studies, we measured the activity of CER synthases 4 and 6, both of them playing essential roles in CER production, especially in resveratrol-treated cells [30]. CER synthase 4 activity was not influenced by resveratrol treatment of A549 cells; only CER synthase 6 was significantly upregulated.

Further studies were carried out on the third enzyme, responsible for the maintenance of CER levels in cells, alkaline ceramidase (ALCER). Its activity was not significantly altered in resveratrol-treated cells (9%), showing that ALCER did not contribute markedly to the elevation in CER levels (Figure 6A). In addition, Western blot analysis did not reveal any significant differences in ALCER expression between resveratrol-treated and untreated lung carcinoma cells (Figure 6B).

### 2.3. Influence of Resveratrol on Sphingosine Kinase 1 Activity and Expression

It is well known that the two most functionally active sphingolipids, CER and S1P, play opposite roles in determining cellular fate: CER acts as a pro-apoptotic agent, whereas S1P stimulates cellular proliferation and survival. The key enzyme that controls the balance between these two sphingolipids is SK1, which phosphorylates SPH to produce S1P. Alterations in SK1 regulation result in changes in the levels of the pro-proliferative lipid S1P, which is accompanied by changes in the balance between apoptosis and survival, and these processes are crucial when it comes to carcinoma cells. Our studies showed that resveratrol downregulated SK1 in A549 cells by 31% (Figure 7A). Western blot analysis showed that resveratrol induced a reduction in sphingosine kinase 1 expression by about 34% (Figure 7B).

Further research was focused on the mechanism of modulation of SK1 by its specific inhibitor, the sphingosine structural analog dimethyl sphingosine (DMS) and the synthetic sphingosine analog FTY720 (2-amino-2-[2-(4-octylphenyl)]-1,3-propanediolhydrochloride), fingolimod, in the presence of resveratrol. For this purpose, A549 cells pretreated with resveratrol were incubated for two hours either with DMS or fingolimod (Figure 8). Incubation of resveratrol-pretreated cells with DMS induced downregulation of SK 1 by 34% and incubation with fingolimod by 27% (Figure 8A). Western blot analysis showed that incubation of resveratrol-pretreated cells with DMS and fingolimod did not cause changes in the enzyme expression, although both of them induced additional downregulation of SK1 activity (Figure 8B), which is an important finding, especially for cancer cells.

## 3. Discussion

Resveratrol (3,5,4′-trihydroxy-trans-stilbene) is a naturally occurring polyphenol that has been reported to possess various beneficial effects, such as anti-inflammatory, anti-tumor, anti-aging, antioxidant, and neuroprotective effects, among others [31]. There is evidence that resveratrol can influence carbohydrate and lipid metabolism in cells. The anti-cancer activity of resveratrol has been related to regulation/dysregulation of sphingolipid metabolism [32,33]. It reduces the production of the pro-survival sphingolipid S1P, inhibits SK1 by influencing phospholipase D, and decreases SK1 expression in prostate cancer cells [34]. Thus, resveratrol suppresses cell proliferation and induces apoptotic processes, most probably by modulating the balance between CER and S1P in cancer cells, this balance being called the “sphingolipid rheostat”. The “sphingolipid rheostat” hypothesis was formulated to explain the controversial physiological activity of the sphingolipids ceramide, sphingosine, and S1P in determining cell fate [35]. According to this hypothesis, the balance between these sphingolipids, which build up specific cascades where one component emerges from the other by sequential enzymatic degradation, determines whether the cells undergo apoptosis or proliferation. CER affects various cellular components, such as PKC, p38, and JNK, that regulate the cell cycle and apoptosis [36]. SPH was reported to inhibit protein kinase C and exhibit an anti-proliferative effect [18,37]. These complex interrelations are crucial for cell biology, and they become targets of different drugs, because the SK/S1P signaling pathway is implicated in the progression of various human diseases, including cancer and inflammation [38,39].

Resveratrol alters sphingolipid levels through multiple mechanisms. For example, it increases CER levels to induce apoptosis in colon, breast, and prostate cancer cells [24,25,26]. There is evidence that resveratrol also downregulates SK1 expression and activity in prostate cancer cells [34]. The inhibition of SK1 activity by resveratrol is likely to influence the balance of the “sphingolipid rheostat”, shifting it to an increase in CER levels. Perturbation of the “sphingolipid rheostat” by resveratrol might, therefore, be responsible to some extent for its anti-cancer effect on cells.

In our previous studies, we reported that resveratrol induced an increase in SM and decrease in CER in senescent rat hepatocytes [40]. In addition, nSMase was downregulated, which we presumed to be a major reason for CER reduction. These findings seem controversial to the results presented in the present paper. However, it is possible that this discrepancy is due to the differences in the types of cells under investigation; in the previous studies, we analyzed normal hepatocytes, isolated from healthy aged animals, whereas, in the present study, we used cancer cells. It is quite likely that resveratrol exhibits diverse effects on CER accumulation depending on the degree of pathophysiological changes occurring in the corresponding cell model. One could further speculate that, in cancer cells, the increase in CER and the consequent initiation of apoptosis is a rational way to eliminate such pathological cells. However, in hepatocytes of old animals, resveratrol reduced the level of CER by inhibiting nSMase activity, which is most probably beneficial for the senescent cells. The effect of resveratrol on SM content is a finding of particular interest. On the one hand, SM is a major component of the membrane raft domains, which are recognized as cellular signaling platforms [41]. On the other hand, SM is the main source of CER, a bioactive second messenger, which is reported to increase in the course of aging and is also considered as a marker of senescence. The accumulation of CER has been correlated with the onset of aging-associated inflammation, cellular senescence, growth arrest, and many aging-associated diseases [42].

The present results showed a decrease in SM in membranes of A549 cells which was not related to the activity of nSMase, the latter remaining almost unchanged (Figure 2A,B). Since neither the activity nor the expression of SMase was influenced by resveratrol treatment, we analyzed the content of SM in the culture medium. Interestingly, the medium contained 21% more SM compared to controls (Figure 3). There is evidence that the enzyme responsible for SM transfer is activated by resveratrol treatment in liver hepatoma cells [27], which could be the reason for the efflux we observed in A549 cells. Since SM exhibits a high affinity for cholesterol (CH) and the two lipids form the so called “raft” domains, the latter acting as signaling platforms in cellular membranes [38], we also analyzed the alteration in the level of CH in resveratrol-treated cells (Figure 3). The content of CH in the incubation medium was elevated by 18% after resveratrol treatment, implying that this phytoalexin probably induces CH efflux, which, together with the membrane depletion of SM, would lead to a decrease in the relative content of raft domains. Such alterations would affect cell signaling and could also be related to an elevation in oxidative stress, because SM functions as an intrinsic antioxidant, preventing the polyunsaturated fatty acids from oxidative destruction [43].

Resveratrol treatment of A549 lung cancer cells induced elevation in CER and SPH and reduction in S1P (Figure 4). CER and SPH are reported to act as pro-apoptotic factors, but they also occur as precursors of the pro-survival S1P. These sphingolipids are interconvertible through a cascade of lipid-metabolizing enzymes. CER levels especially are maintained through the balance of three sphingolipid-metabolizing enzymes: SMase, CER synthase, and ceramidase. Our studies showed that resveratrol treatment did not induce significant changes, neither in the activity nor in the expression of nSMase in A549 cells (Figure 2), which rules out the participation of the nSMase pathway in the elevation in CER levels. Research was focused on CER synthase 6, which is a key enzyme in the CER anabolic pathway [30]. CER synthase 6 was upregulated by 38% in resveratrol-treated A549 cells compared to untreated ones (Figure 5A), implying that this enzyme activity is likely to underlie the observed elevation in CER. In addition, Western blot analysis showed that CER synthase 6 expression was upregulated by 32% in the treated cells. These data clearly show that resveratrol affected both the activity and expression of CER synthase 6, which resulted in significant augmentation in the membrane levels of the pro-apoptotic sphingolipid CER in lung cancer cells.

Another possible source of CER augmentation could be downregulation of ALCER. As evident from Figure 6, ALCER was upregulated by about 9% as a result of resveratrol treatment, implying that this enzyme could not be responsible for the elevation in CER, because it hydrolyzes CER to produce SPH. In addition, ALCER expression was not affected significantly by resveratrol treatment of A549 adenocarcinoma cells. However, the upregulation of ALCER, although very weak, could be partially responsible for the augmented levels of SPH, possibly due to direct interaction of resveratrol with ALCER or with components of its metabolic pathway. However, the expression of this enzyme remained unchanged (Figure 6B). The elevation in SPH levels was an important observation, because SPH is a pro-apoptotic factor, but, more importantly, it acts as a precursor of S1P, which is a pro-survival sphingolipid with a very high physiological activity. As mentioned above, the balance between SPH and S1P is maintained by a key enzyme in the sphingolipid pathway, SK1. The activity of SK1 is essential for the determination of cell fate, and approaches to influencing its regulation attract great research interest, especially when it comes to cancer cells. Since S1P was reduced in resveratrol-treated cells (Figure 4), we analyzed the activity and level of expression of SK1 in control and resveratrol-treated cells (Figure 7A,B). SK1 was downregulated by 31% in the treated cancer cells, and Western blot analysis showed a decrease in expression by 28%. Thus, resveratrol affects the activity of almost all catabolic enzymes in the SM metabolic pathway in A549 lung adenocarcinoma cells and alters the expression of two key lipid metabolizing enzymes, CER synthase 6 and SK1. Since SK1 is essential for the accumulation of S1P and both its activity and expression are altered by resveratrol, we further analyzed the mechanism of influence of this phytoalexin on SK1 activity in the presence of two effectors which are structural analogs of SPH: the specific SK1 inhibitor DMS and the synthetic sphingosine analog fingolimod, which are competitive inhibitors of SK1. In our previous studies, we reported a synergistic effect of DMS together with the anti-tumor agent miltefosine on A549 cells [44]. Both of these agents are related to sphingolipid metabolism, and their synergistic cytotoxic effect is associated with SK1 and the levels of S1P. In the present studies, we analyzed the mechanism of action of resveratrol in combination with two specific SK1 inhibitors (Figure 8). We chose to use the competitive inhibitor of SK1, DMS, which induces a reduction in the pro-survival agent S1P. Of the two identified SKs, sphingosine kinase type 1 has been shown to regulate various processes important for cancer progression [45]. Inhibition of SK1 results in a decrease in the proliferative potential of cells at the expense of apoptosis [46].

FTY720 is a competitive sphingosine inhibitor of SK1 that has been reported to stimulate relocalization of actin away from the lamellipodia of breast cancer cells, suggesting its possible application for the prevention of tumor metastasis [47,48]. Interestingly, FTY720 is approved for use as an immunosuppressant in patients with multiple sclerosis [49]. However, the mechanisms underlying the receptor-dependent and -independent functions of FTY720 in the context of its reported inhibitory effects on cancer cell proliferation are largely unclear. The combined treatment of A549 cells with resveratrol and DMS induced an additional downregulation of SK1, and, although they did not affect SK1 expression, both effectors acted in the same direction in reducing the level of S1P and, consequently, the survival potential of the treated cells. (Figure 8A,B). Similarly, the combined effect of resveratrol and FTY720 on SK1 also showed an additive effect on SK1 activity, both inducing a reduction in the pro-survival factor S1P without influencing the enzyme expression. The presented results show that the pro-apoptotic effect of resveratrol on lung cancer A549 cells can be increased by combining it with other biologically active agents which influence the activity of SK1.

In conclusion, the present studies provide information concerning the biochemical processes underlying the influence of resveratrol on sphingolipid metabolism in A549 lung cancer cells (Figure 9) and reveal possibilities for the combined use of polyphenols with specific anti-proliferative agents. Such studies could serve as the basis for development of complex therapeutic strategies involving polyphenols, which are natural products with various beneficial properties.

## 4. Materials and Methods

### 4.1. Reagents

Resveratrol was purchased from Sigma–Aldrich. DMEM and penicillin-streptomycin (10,000 U·mL^−1^ penicillin and 10,000 mg·mL^−1^ streptomycin) were from Invitrogen (Paisley, UK). DMSO and MTT were from MP Biomedicals, LLC. SPH and S1P were from Avanti Polar Lipids (Alabaster, AL, USA). D-erythro-N,N-Dimethylsphingosine (DMS) was purchased from Santa Cruz Biotechnology, USA. It was dissolved in DMSO and was stored at −20 °C until use. Fingolimod was from Cayman Chemicals (Ann Arbor, MI, USA Cat # 11975).

### 4.2. Cell Culture and Incubation with Resveratrol

Human lung carcinoma cells A549 (ATCC No CCL-185) were grown in DMEM, 10% FBS (HyClone), 100 Units/mL ampicillin, and 100 µg/mL streptomycin (Sigma) at 37° in a humidified atmosphere containing 5% CO_2_. In these experiments, we used preparations which exhibited at least 90% viability. The A549 cells (5 × 10^5^/well) were seeded in a 24-well plate overnight and treated with 100 µM of resveratrol for 24 h. This concentration of resveratrol was chosen because the measured alterations in sphingomyelin levels were linear up to 150 µM. Resveratrol was delivered from a stock solution in dimethyl sulfoxide. Control cells were incubated only with dimethyl sulfoxide. After 24 h, the cells were incubated with either 100 µM DMS or 50 µM fingolimod for two more hours.

### 4.3. Cell Viability Assay after Incubation with Resveratrol

After incubation with resveratrol, cell viability was determined by tetrazolium salt measurement (MTT assay), involving the assessment of succinate dehydrogenase-induced conversion of (3-[4,5-dimethylthiazol-2-yl]-2,5-diphenyl tetrazolium bromide into formazan crystals. Formation of formazan was measured at 570 nm. The viability of the cells after incubations was estimated as a percentage of the absorbance of resveratrol-treated cells compared to controls.

### 4.4. Isolation of Plasma Membrane from A549 Cells

The plasma membrane fraction was obtained according to the procedure described by Evans [50] with modifications, including differential centrifugation of A549 cells. The post-nuclear supernatant was loaded on a discontinuous sucrose gradient and centrifuged at 100,000× *g* for 2.5 h. The plasma membrane fraction was collected at a density of approximately 45% (*w*/*v*), suspended in ice-cold 100 mM Tris buffer, pH 7.5, and used immediately for lipid analysis

### 4.5. Lipid Analysis

Cholesterol content was assayed by gas chromatography using a medium polarity RTX-65 capillary column (0.32 mm internal diameter, length 30 m, thickness 0.25 µm). Calibration was achieved by a weighted standard of cholestane.

### 4.6. Determination of Sphingomyelin

This analysis was performed with the Sphingomyelin Quantification assay kit (Sigma-Aldrich), Cat.# MAK262. The assay is based on the hydrolysis of sphingomyelin to ceramide and phosphocholine by sphingomyelinase. Alkaline phosphatase (ALP) dephosphorylates phosphocholine to choline, producing a reaction that generates a colorimetric signal. After treatment of cells, the supernatant was collected, centrifuged to remove cell debris, and kept for analysis. The concentration of SM in a sample was calculated as nmol SM per μL

### 4.7. Determination of Ceramide

The content of ceramide was determined using the Ceramide ELISA kit (MyBioSource, Cat.# 3801246) according to the manufacturer’s instructions. The stop solution changed the color from blue to yellow, and the intensity of the color was measured at 450 nm. Ceramide concentration was calculated using a standard curve which was generated by plotting the average optical density obtained for each of the standard concentrations on the vertical (Y) axis versus the corresponding concentration on the horizontal (X) axis.

### 4.8. Determination of Sphingosine

Sphingosine level was determined using the sphingosine ELISA kit (Aviva Systems Biology, Cat.# OKEH02615) according to the manufacturer’s instructions. This assay is based on the competitive enzyme immunoassay technique. After the final enzymatic reaction, the density of yellow coloration was measured by reading the absorbance at 450 nm, which is quantitatively proportional to the amount of biotinylated sphingosine. A standard curve was generated by plotting the mean optical density at 450 nm vs. the respective standard concentration.

### 4.9. Determination of Sphingosine-1-Phosphate (S1P)

S1P was determined using the ELISA kit for detecting the amount of human sphingosine-1-phosphate in biological samples (Cat.# 682861, Antibody Research Corporation, USA) based on the principle of the double-antibody sandwich technique. After treatment of cells, the supernatant was collected, centrifuged, and kept for analysis. Then, the procedure described by the manufacturer was followed, and the optical density of the samples was measured at 450 nm. The amount of S1P in the samples was calculated using a standard curve.

### 4.10. Neutral Sphingomyelinase (nSM) Activity Assay

The activity of nSM was measured by the Neutral Sphingomyelinase Activity assay kit (Echelon Biosciences, Cat.# K-1800). In this enzyme-coupled assay, n-SMase catalyzes the hydrolysis of sphingomyelin into phosphorylcholine and ceramide. Alkaline phosphatase then catalyzes phosphorylcholine to choline. Choline is oxidized by choline oxidase to form hydrogen peroxide. Finally, 4-aminoantipyrine (4 AAP) in the presence of hydrogen peroxide and peroxidase results in the oxidation of 4 AAP to form a blue chromogen that is detected by measuring the absorbance of light at 595 nm.

### 4.11. Ceramide Synthase 6 Activity Assay

Ceramide synthase activity was determined using the Mouse Ceramide Synthase 6 (Cer6) ELISA kit (Abbexa, Cat.# abx506301). This enzyme catalyzes the acylation of sphingosine to yield ceramide. This kit is based on sandwich enzyme-linked immuno-sorbent assay technology. The HRP-conjugated reagent was added, and the whole plate was incubated. Unbound conjugates were removed using wash buffer at each stage. After the substrate was added, only wells that contained sufficient CER synthase 6 produced a blue colored product, which changed to yellow after adding the acidic stop solution. The intensity of the yellow color is proportional to the amount of CER synthase 6 bound to the plate. The optical density was measured spectrophotometrically at 450 nm in a microplate reader, from which the concentration of CER synthase 6 was calculated.

### 4.12. Alkaline Ceramidase (ALCER) Activity Assay

Alkaline ceramidase activity was determined by measuring the released sphingosine from ceramide [39]. Briefly, A549 cells (75 µg protein) were incubated with ceramide substrate (250 µM) in 25 mM Tris-HCl buffer, pH 9, containing 5 mM CaCl_2_ and 5 mM MgCl_2_ at 37 °C for 30 min. The reaction was stopped by the addition of 0.5 mL CHCl_3_/CH_3_OH (2:1, *v*/*v*). The produced sphingosine was determined by the sphingosine ELISA kit (Aviva Systems Biology, Cat.# OKEH02615).

### 4.13. Sphingosine Kinase 1 Activity Assay

Sphingosine kinase 1 activity was determined using the sphingosine kinase 1 assay ELISA kit (SPK1 DuoSet ELISA, R&D Systems, Cat.# DY5536) as recommended by the manufacturer’s instructions. Optical density was measured immediately after the addition of stop solution to each well, using a microplate reader at 450 nm.

### 4.14. Western Blotting and Antibodies

Control and resveratrol-treated A549 cultures were solubilized in RIPA buffer (150 mM NaCl, 2 mM EDTA, 1% sodium deoxycholate, 0.1% SDS, 1% Triton X-100, 10% glycerol, 50 mM HEPES, pH 7.5) containing Complete Inhibitory Cocktail (Cat.#11697498001, Merck). Equal volumes of 5x sample buffer (60 mM Tris-HCl, pH 6.8, 2% SDS, 10% glycerol, 5% b-mercaptoethanol and bromophenol blue) were added, and samples were heated for 4 min at 95 °C. The three replicate samples from control and resveratrol-treated cells, respectively, from each individual experiment were combined and run on a single lane. Proteins were resolved on SDS-PAGE, transferred to nitrocellulose membrane, and blocked for 1 h in 5% non-fat dry milk in TBST (50 mM Tris base, 200 mM NaCl, 0.1% Tween-20, pH 7.4). Membranes were then incubated with the appropriate primary and secondary antibodies, including: monoclonal anti-neutral sphingomyelinase 2 (Cat.# sc-166637), ceramide synthase 6 (Cat.# sc-100554, LASS6), anti-sphingosine kinase 1 (Cat.# sc-365401), anti-GAPDH (Cat.# sc-32233) (all from Santa Cruz Biotechnology), anti-alkaline ceramidase 2 (Cat.# A09706-1, Boster Biological Technology), anti-mouse IgG (Fab) HRP conjugate (Cat.# SAB-100, Stressgen), and anti-rabbit IgG (Fab) HRP conjugate (Cat.# SAB-300, Stressgen). Immunoblots were visualized using the ECL system (Santa Cruz).

### 4.15. Statistical Analysis

Statistical processing of the data was performed with the unpaired *t*-test (for the results presented in Figure 1, Figure 2, Figure 3, Figure 4, Figure 5, Figure 6 and Figure 7) and one-way analysis of variance (ANOVA) for the results presented in Figure 8, using In Stat software (Graph Pad In Stat 3.1, developed by Graph Pad Software, San Diego, CA, USA).

The analyzed data for each figure were obtained from three independent experiments, i.e., from experiments performed on three different days, each of them containing three replicates, finally resulting in a total of nine cell lysates for the control and as many for each treatment. In immunoblotting experiments, lysates from each independent experiment were pooled so that all preparations could be loaded onto a single gel and treated under the same antibody conditions.

All data related to the investigated amounts of sphingolipids and enzyme activities passed a normality test. The results from the immunoblottings where pooled samples were used did not provide a sufficient number of data points to perform this test. Therefore, the specific values from each independent experiment are indicated in the corresponding figure.

## 5. Conclusions

Resveratrol treatment of A549 cells induced an increase in ceramide and sphingosine and decrease in sphingomyelin and sphingosine-1-phosphate.Sphingomyelinase was not responsible for sphingomyelin reduction, but an elevation in sphingomyelin was observed in the incubation medium after resveratrol treatment.Resveratrol induced upregulation and elevated expression of ceramide synthase in A549 cells.Sphingosine kinase 1 was downregulated, and its expression was reduced in resveratrol treated A549 cells.Incubation of A549 with the SK1 inhibitors DMS and fingolimod induced additional downregulation of SK1 in resveratrol-treated lung adenocarcinoma cells, implying that the combined use of anti-proliferative effectors with naturally occurring anti-tumor agents could turn out to be a useful tool for reducing the pro-survival potential of cancer cells.

## Figures and Tables

**Figure 1 ijms-23-10870-f001:**
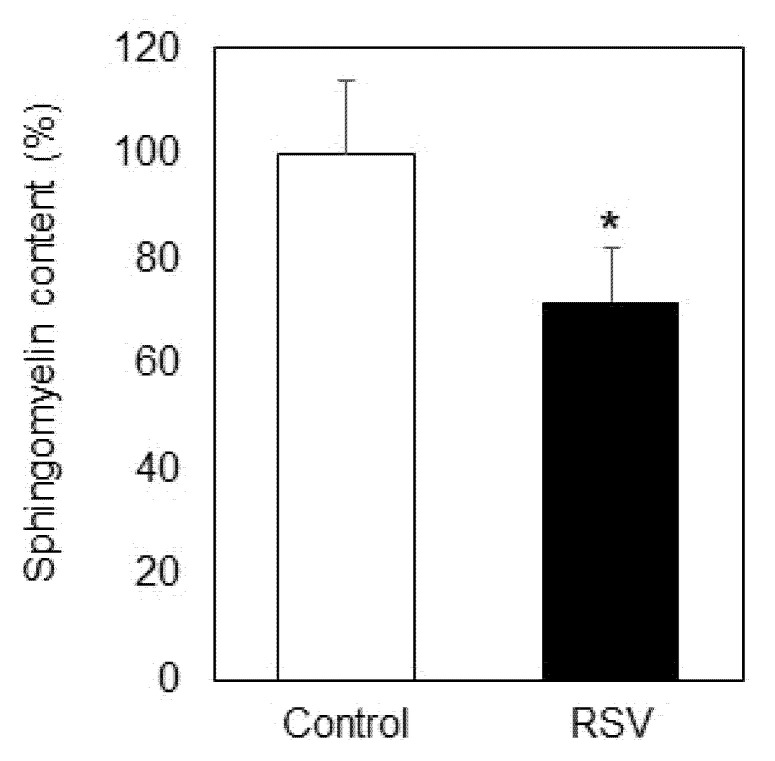
Sphingomyelin (SM) content in plasma membranes isolated from A549 cells untreated (Control, white bars) and treated with resveratrol (RSV, black bars). Values are expressed as relative percentage participation in the total lipids. Values are means ± SD. * *p* < 0.01.

**Figure 2 ijms-23-10870-f002:**
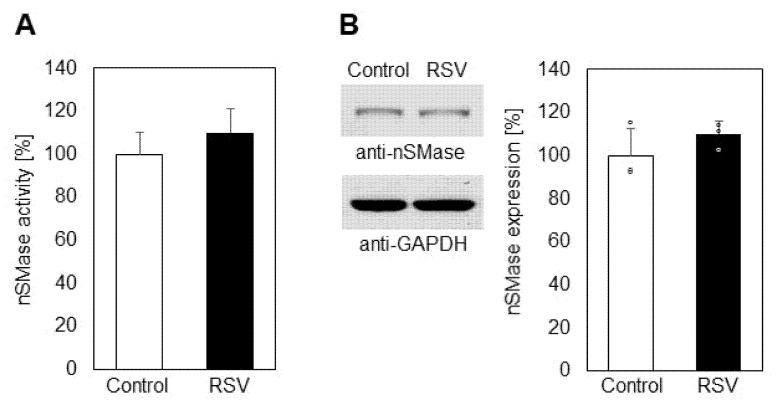
Changes in the activity (**A**) and protein expression (**B**) of neutral sphingomyelinase (nSMase) in control (white bars) and resveratrol-treated (black bars) A549 cells. Values are expressed as % change of controls (100%). Representative images from Western blot analysis with specific antibodies to neutral sphingomyelinase (nSMase) are shown in the left part of panel B. Reaction with anti-glyceraldehyde-3-phosphate dehydrogenase antibodies (anti-GAPDH) was used as an internal control for loading; (ᵒ) stands for the values from each independent experiment, obtained by pooling of three replicate samples. Values represent means ± SD. The differences between the obtained values were not statistically significant.

**Figure 3 ijms-23-10870-f003:**
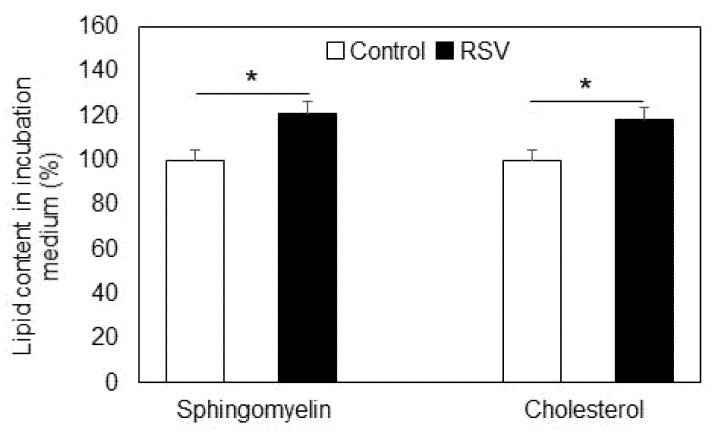
Content of sphingomyelin and cholesterol in the incubation medium of A549 cells untreated (Control, white bars) and treated with resveratrol (RSV, black bars). Values are expressed as % of controls (100%). Values are means ± SD. The differences between the values obtained for both SM and CH were statistically significant. * *p* < 0.01.

**Figure 4 ijms-23-10870-f004:**
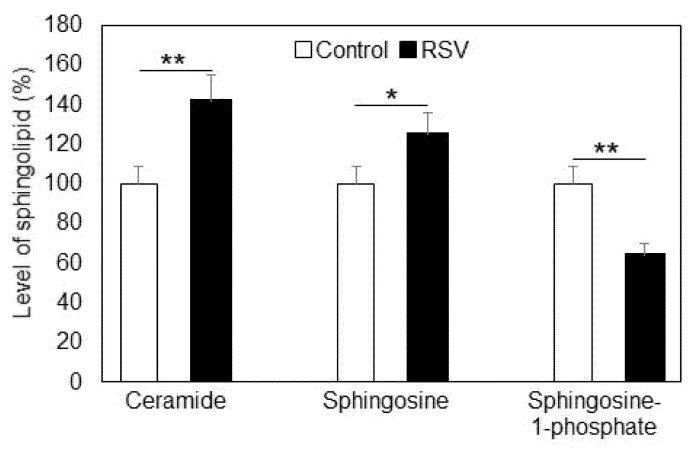
Changes in the level of ceramide, sphingosine, and sphingosine-1-phosphate in control (Control, white bars) and resveratrol-treated (RSV, black bars) A549 cells. Values are expressed as % of controls (100%). Values are means ± SD. The differences between the values obtained for ceramide (** *p* < 0.001), sphingosine (* *p* < 0.01), and sphingosine-1-phosphate (** *p* < 0.001) were statistically significant.

**Figure 5 ijms-23-10870-f005:**
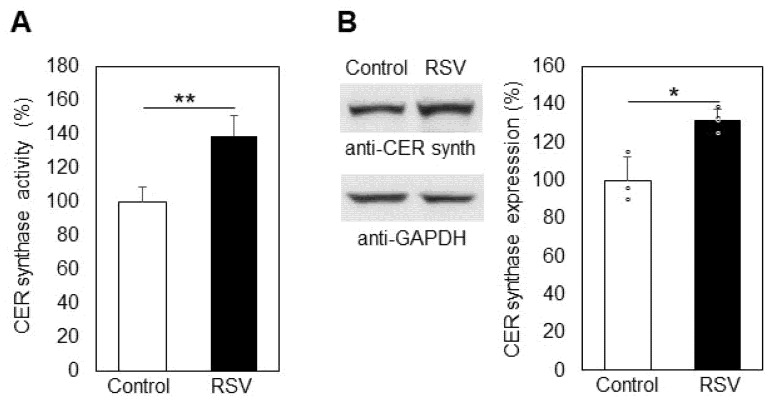
Alteration in the activity (**A**) and the expression (**B**) of ceramide synthase 6 in control (Control, white bars) and resveratrol-treated (RSV, black bars) A549 cells. Values are expressed as % of controls (100%). Representative images from immunoblotting obtained with antibodies against ceramide synthase (anti-CER synth) are shown on the left part of panel B. Reaction with anti-glyceraldehyde-3-phosphate dehydrogenase antibodies (anti-GAPDH) was used as an internal control for loading. Graphical depiction of the percent change in CER synthase expression is presented on the right part of panel B; (ᵒ) stands for the values from each independent experiment, obtained by pooling of three replicate samples. Values represent means ± S.D. * *p* < 0.01, ** *p* < 0.001.

**Figure 6 ijms-23-10870-f006:**
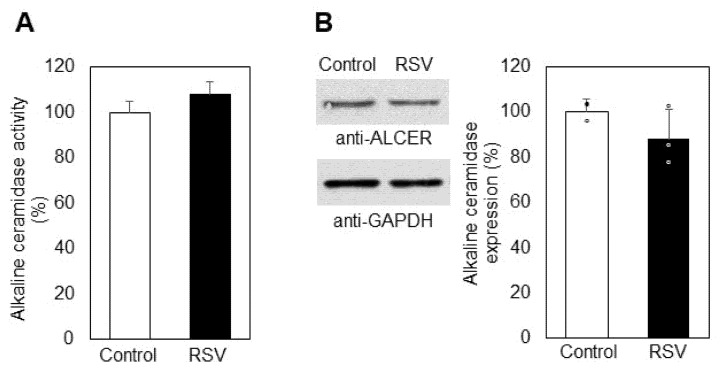
Specific activity (**A**) and protein expression (**B**) of alkaline ceramidase in control (Control, white bars) and resveratrol-treated (RSV, black bars) A549 cells. Values are expressed as % change of control (100%). Representative images from Western blot analysis obtained with antibodies against alkaline ceramidase (anti-ALCER) are shown on the left part of panel B. Reaction with anti-glyceraldehyde-3-phosphate dehydrogenase antibodies (anti-GAPDH) was used as an internal control for loading. Graphical depiction of the percent change in alkaline ceramidase expression is presented on the right part of panel B; (ᵒ) stands for the values from each independent experiment, obtained by pooling of three replicate samples. Values are means ± SD. The differences between the obtained values were not statistically significant.

**Figure 7 ijms-23-10870-f007:**
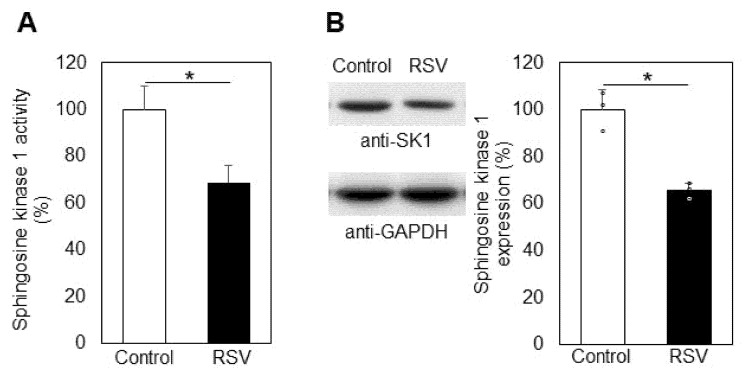
Inhibition of the activity (**A**) and the expression (**B**) of sphingosine kinase 1 in control (Control, white bars) and resveratrol-treated (RSV, black bars) A549 cells. Values are expressed as % of controls (100%). Representative images from Western blot analysis obtained with antibodies against sphingosine kinase 1 (anti-SK1) are shown on the left part of panel B. Reaction with anti-glyceraldehyde-3-phosphate dehydrogenase antibodies (anti-GAPDH) was used as control for loading. Graphical depiction of the percent change in SK1 expression is presented on the right part of panel B; (ᵒ) stands for the values from each independent experiment, obtained by pooling of three replicate samples. Values represent means ± S.D. * *p* < 0.01.

**Figure 8 ijms-23-10870-f008:**
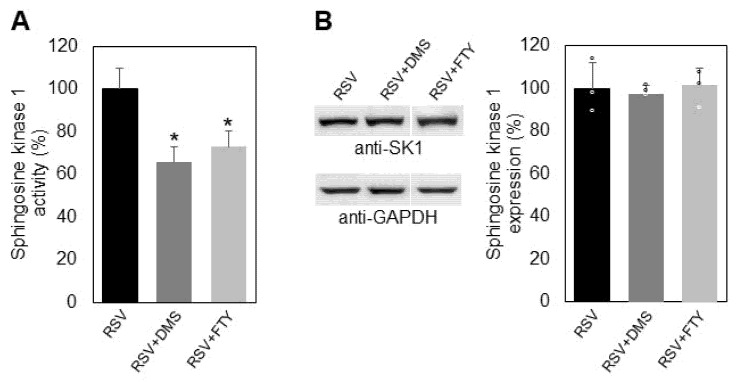
Alterations in the activity (**A**) and expression (**B**) of sphingosine kinase 1 in A549 cells incubated only with resveratrol (RSV, black bars), resveratrol and DMS (RSV+DMS, dark gray bars), and resveratrol plus fingolimod (RSV+FTY, light gray bars). Illustrative images from Western blots, obtained with antibodies against sphingosine kinase 1 (anti-SK1), are shown on the left part of panel B. Reaction with anti-glyceraldehyde-3-phosphate dehydrogenase antibodies (anti-GAPDH) was used as internal control for loading. Graphical representation of the percent change in SK1 expression is presented on the right part of panel B. Values are expressed as % of RSV-treated A549 cells serving as controls; (ᵒ) stands for the values from each independent experiment, obtained by pooling of three replicate samples. Values are means ± SD. * *p* < 0.01.

**Figure 9 ijms-23-10870-f009:**
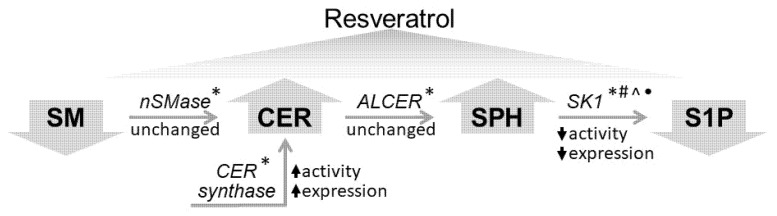
Schematic presentation of the effects of resveratrol on sphingolipid metabolism in lung cancer A549 cells. SM—sphingomyelin; CER—ceramide; SPH—sphingosine; S1P—sphingosine-1-phosphate; nSMase—neutral sphingomyelinase; ALCER—alkaline ceramidase; SK1—sphingosine kinase 1; CER synthase—ceramide synthase. (*) indicates the presence in the literature of information about resveratrol-induced changes in HepG2 cells [30]; (#)—in breast cancer cells [27]; (^)—in prostate cancer cells [28]; and (•)—in a mouse model of acute peritonitis [29].

## Data Availability

Data is contained within the article.

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
