# Peer review of "Resveratrol Affects Sphingolipid Metabolism in A549 Lung Adenocarcinoma Cells"

_ijms, 2022, doi:10.3390/ijms231810870_

Round 1
Reviewer 1 Report
The studies, reported "Resveratrol affects sphingolipid metabolism in A549 lung adenocarcinoma cells" by Prof. Albena Momchilova et. al. provide information concerning the mechanism underlying the influence of resveratrol on the sphingolipid metabolism in A549 cell line. The text is written well (although there are quite many typos or small mistakes). The data is consistent and, as far as I can see, in agreement with the published papers in the field.
Unfortunately, I'm very concerned about reproducibility of the results and the statistics. That is, as I can judge from the blots and figure legends, everything is based on the results obtained "from at least three independent experiments". The individual values are not shown on the graphs (which is very important for the results reporting statistically small samples). Statistical analysis is badly described. It's not stated, whether the normality of the data was checked to use ANOVA. Moreover, ANOVA can be used for at least 3 groups (as in Fig. 8), but the figures 1-7 report data from just two groups (then Mann-Whitney test should be used, however it is useless for n<3 too).
Thus, 1) clarification of statistical analysis is of utmost importance.
2) The data needs reproduction as independently seeded cell cultures (all technically repeated in triplicates). The term "independent experiments" should be avoided or explained in details.
3) Since the direct effect of resveratrol on sphingosine kinase 1 is known for more than a decade, the quality of the paper would be significantly improved if one or two more cell types were used. One could compare the effect of resveratrol on A549 and for example SCLC-21H lung cancer cells. This is not obligatory, but would improve the scientific importance of the reported results, especially since the authors try to expand their observations to all "lung cancer cells".
4) The data should be compared to the effect of resveratrol on normal cells as a comparison to cancer cells. The authors mention a contradiction between their previous studies on rat hepatocytes and these data. Here a comparison to normal lung cells would be the best variant. The mentioned contradiction also supports the statement about reproducibility.
5) It would be good to add a scheme (for the discussion) or graphical abstract.
Unfortunately, the reported results can't be accepted for publication, with the high risk of non-reproducibility as the major problem.
Author Response
We would like to thank this reviewer for his/her recommendations, which, no doubt, would contribute to making this paper an interesting source of information.
We have taken into consideration every point raised in this review and have made the required changes as follows:
1) Reviewer: clarification of statistical analysis is of utmost importance.
Answer: We thank the reviewer for the relevant question regarding the description of the statistical methods we used. In the submitted manuscript, we omitted to note that in experiments comparing control and treated cells we used unpaired t-test, and ANOVA was used only to assess the reliability of the results obtained with the sphingosine kinase 1 inhibitors. In the revised manuscript, this information is presented in the Statistical analysis section:
„Statistical processing of the data was made by unpaired t-test (for the results presented on figures 1 to 7) and one-way analysis of variance (ANOVA) for results presented on Fig. 8, using In Stat software Graph Pad In Stat 3.1, developed by Graph Pad Software, San Diego, CA, USA
2) Reviewer: The data needs reproduction as independently seeded cell cultures (all technically repeated in triplicates). The term "independent experiments" should be avoided or explained in details.
Answer: We fully agree that sufficient replications of any experiment are extremely important for the reliability of the results. In our studies, the term "independent experiment" is used to refer to an experiment performed on a particular day in which a population of trypsinized cells is seeded at equal concentration in three Petri dishes for the respective treatment and a similar number of Petri dishes for the control. This means that for each independent experiment three preparations of the control and three preparations of the resveratrol treatment were used. The figures presented in the manuscript were obtained from three independent experiments, e.g. from experiments performed on three different days with a total of nine cell lysates for the control and as many for each treatment. In immunoblotting experiments, lysates from each independent experiment were pooled so that all preparations could be loaded onto a single gel and treated under the same antibody conditions. For this reason, the presented gels have three runs for the control and three runs for the resveratrol treatment, but each run contains lysates from three petri dishes. To clarify this issue we have included in the Statistical Analysis section the following:
“The analyzed data were obtained from three independent experiments performed on different days and containing three replicates for the control and each of the treatments.”
Pooling of the preparations is explained in the Western blotting and antibodies section as follows:
“The three replicate samples from control and resveratrol-treated cells respectively from each individual experiment were combined and run on a single lane”
3) Reviewer: Since the direct effect of resveratrol on sphingosine kinase 1 is known for more than a decade, the quality of the paper would be significantly improved if one or two more cell types were used. One could compare the effect of resveratrol on A549 and for example SCLC-21H lung cancer cells. This is not obligatory, but would improve the scientific importance of the reported results, especially since the authors try to expand their observations to all "lung cancer cells".
Answer: We absolutely agree that the direct effect of resveratrol on sphingosine kinase 1 has been observed years ago. For example, such effect was reported for leukemia cell line K562 (Resveratrol induces apoptosis of leukemia cell line K562 by modulation of sphingosine kinase-1 pathway. Tian H, et al. Int J Clin Exp Pathol. 2015), MCF-7 breast cancer cells (Resveratrol dimers are novel sphingosine kinase 1 inhibitors and affect sphingosine kinase 1 expression and cancer cell growth and survival. Lim KG, et al. Br J Pharmacol. 2012), Hep G2 cells (Charytoniuk, T.; Harasim-Symbor, E.; Polak, A. et al. Influence of Resveratrol on Sphingolipid Metabolism in Hepatocellular Carcinoma Cells in Lipid Overload State, Anti-Cancer Agents in Medicinal Chemistry, 2019, 19, 121-129) etc. Especially for A549 cells there is evidence that resveratrol causes increase of apoptosis and cell cycle arrest through upregulation of of p53 and p21 nuclear expression, reduced cell proliferation and cell growth, lysosomal membrane permeabilization, up-regulation and cleavage of caspase 3, etc. However, to our knowledge, there is no analysis of the activity and expression of the sphingolipid metabolites and especially of sphingosine kinase 1 induced by resveratrol and certain sphingosine structural analogs in A549cells, which is why we performed these studies. In addition, we have specified in the text that the reported results refer to A549 cells, and not to lung cancer cells in general, as recommended by the reviewer (Abstract, page 12, page 15).
4) Reviewer: The data should be compared to the effect of resveratrol on normal cells as a comparison to cancer cells. The authors mention a contradiction between their previous studies on rat hepatocytes and these data. Here a comparison to normal lung cells would be the best variant. The mentioned contradiction also supports the statement about reproducibility.
Answer: The different (diverse) effect of resveratrol on normal and cancer cells has been well documented by other authors (Lu J, Ho CH, Ghai G, Chen KY. Resveratrol analog, 3,4,5,4-tetrahydroxystilbene, differentially induces proapoptotic p53/Bax gene expression and inhibits the growth of transformed cells but not their normal counterparts. Carcinogenesis 2001;22:321 – 8). Our studies also confirmed this different effect, although the cells we analyzed were not similar. The point is that when applied to healthy (non-cancer) cells, which were senescent hepatocytes in our previous studies (Momchilova, A. et al. Resveratrol alters the lipid composition, metabolism and peroxide level in senescent rat hepatocytes. Chem. Biol. Interact. 2014, 207, 74-80), resveratrol does not induce apoptosis, but it contributes to reduction of the pro-apoptotic factor ceramide. We also consider as a very important result of resveratrol treatment the observed in our lab elevation of sphingomyelin level in non-cancerous cells, which stabilizes the membrane raft domains and also acts as intrinsic membrane antioxidant. However, in the cancer cells we observed the opposite effect – reduction of sphingomyelin and increase of the pro-apoptotic ceramide, which is a significant difference, especially when it comes to tumor cells. The mentioned above data may seem controversial at first sight, but they only support the notion that resveratrol exhibits diverse effects when applied on normal and cancer cells, this making it an adequate therapeutic tool which induces apoptotic processes in cancer cells and preserves normal ones.
5) Reviewer: It would be good to add a scheme (for the discussion) or graphical abstract.
Answer: We have added in the Discussion section a scheme (Fig. 9), which illustrates our observations of resveratrol effect on sphingolipid metabolism in A549 cells.
Once again I would like to thank this reviewer for the valuable recommendations.
Reviewer 2 Report
The Authours study the effect of resveratrol on sphingolipid metabolism in A549 lung carcinoma cell to investigate the possible therapeutic use of natural substances. This is an important research field because there is a growing need for new therapeutic streategies and the sphingolipid metabolism could be a promising and attractive target of new anticancer drugs.
The Authors have produced a great effort, but in my opinion the paper needs some changes and implementations.
In the introduction it was not discussed how lipid metabolism has already been studied and investigated in lung cancer. It would be useful to deepen this aspect with the most recent references.
The Authors reported only the results for the concentration of 100micromolar and justified their choice, but more interesting would be to undestand if there are some trends in the expression of the investigated enzymes and lipids in resveratrol treated cells.
Even if the enzymes expression/content and content of the studied sphingolipid are studied, there are not specific experiments about the effects on cell cycle, proliferation, apoptosis and maybe morpholgy of cells treated with resveratrol. I think these experiments are necessary.
Moreover the use of the term "mechanism" when the authors report the results of SK1 inhibition by DMS and FTY 720 in resveratrol-pretreated cells is not in my opinion the most appropriate, as the authors only evaluate the changes in expression and activity levels of this enzyme.
Why at line 152 there is "Table 0. and sphingosine-1-phosphate (**P<0.001) were statistically significant"?
Please indicate the inhibitor FTY 720 in the same way always in the paper (there are FTY720, FTY 720 and FTY-720)
Please indicate resveratrol as 3,5,4′-trihydroxy-trans-stilbene at the first citation in the text and not in the discussion
In conclusion I think the paper could deserve the pubblication after major revision
Author Response
I would like to thank this reviewer for the useful remarks, which, no doubt, would improve the quality of this paper and would make it more readable and comprehensive.
We have taken into consideration every point raised in this review and have made the required changes as follows:
Reviewer: In the introduction it was not discussed how lipid metabolism has already been studied and investigated in lung cancer. It would be useful to deepen this aspect with the most recent references.
Answer: As suggested by this Reviewer we have added to the Introduction section references # 21, 22, 23, which are related to the sphingolipid metabolism in lung cancer cells. However, to our knowledge there are no studies devoted to the effect of resveratrol on the sphingolipid metabolism in A549 cells.
Reviewer: The Authors reported only the results for the concentration of 100 micromolar and justified their choice, but more interesting would be to undestand if there are some trends in the expression of the investigated enzymes and lipids in resveratrol treated cells.
Answer: We have explained the reason for the choice of resveratrol concentration in the present studies. In previous experiments we have incubated other cancer cells (Hep G2 and ras-transformed fibroblasts) with resveratrol concentration different form 100 µM and have observed similar tendencies, especially for elevation of ceramide level and down-regulation of sphingosine kinase 1. However, for ras-transformed fibroblasts the augmentation of ceramide was not due to up-regulation of ceramide synthase as observed for A549 cells, but rather to up-regulation of neutral sphingomyelinase. These results are being under review for another journal which is why they have not been included in the present paper. Still, there was a common tendency between the influence of resveratrol on A549 cells and ras-transformed fibroblasts and that is the reduction of sphingomyelin and the elevation of ceramide, which seem to be a result of different biochemical processes for the two cell lines and these observations deserve special attention.
Reviewer: Even if the enzymes expression/content and content of the studied sphingolipid are studied, there are not specific experiments about the effects on cell cycle, proliferation, apoptosis and maybe morphology of cells treated with resveratrol. I think these experiments are necessary.
Answer: In these experiments we have analyzed the alterations in the sphingolipid metabolism and metabolites induced by resveratrol treatment, as stated in the title. Studies on the alterations in apoptosis and proliferation are in progress in our lab and will be subject of another paper devoted to the degree of apoptosis, necrosis and proliferation induced by resveratrol, alone and in combination with specific sphingolipid-related effectors.
Reviewer: Moreover the use of the term "mechanism" when the authors report the results of SK1 inhibition by DMS and FTY 720 in resveratrol-pretreated cells is not in my opinion the most appropriate, as the authors only evaluate the changes in expression and activity levels of this enzyme.
Answer: We have substituted the term “mechanism” with “processes”, as recommended by the reviewer (in the Abstract and in the end of the Discussion section).
Reviewer: Why at line 152 there is "Table 0. and sphingosine-1-phosphate (**P<0.001) were statistically significant"?
Answer: We are very sorry but we could not find at line 152 “Table 0”. Moreover, there are no tables in this paper, only figures. Again, we apologize for not being able to answer to this comment.
Reviewer: Please indicate the inhibitor FTY 720 in the same way always in the paper (there are FTY720, FTY 720 and FTY-720)
Answer: We thank the Reviewer for noticing this incorrectness. We have corrected all variants to FTY720 (page 15).
Reviewer: Please indicate resveratrol as 3,5,4′-trihydroxy-trans-stilbene at the first citation in the text and not in the discussion
Answer: Again, we would like to thank this Reviewer for pointing out this omission. We have corrected that in the Introduction of the amended version.
Finally, I would like to thank this reviewer for giving us valuable advices and suggestions, which would, undoubtedly, improve the quality of the paper.
Round 2
Reviewer 1 Report
Dear authors,
I'm very glad with the improvements and most of your answers.
A few rather small improvements or clarifications are needed.
First of all, you didn't check normality of the data. Could you state clearly, if the data could pass a normality test or not? This is of utmost importance, when you use t-test.
Most of the data, including graphs, are based on the comparison of sets of three data points. Without a statement about normality (or outliers) or adding the exact points on each graph, there still exist a possibility, that some statistical results might be concluded from the effect of e.g. outliers.
You wrote in the answers: "The figures presented in the manuscript were obtained from three independent experiments, e.g. from experiments performed on three different days with a total of nine cell lysates for the control and as many for each treatment. In immunoblotting experiments, lysates from each independent experiment were pooled so that all preparations could be loaded onto a single gel and treated under the same antibody conditions.".
This explanation, containing a statement about 9 cell lysates (and their pooling) is better, than the current description, which contains "three replicate samples" and "data were obtained from three independent experiments performed on different days and containing three replicates for the control and each of the treatments". Could you indicate of the 9 cell lysates in the manuscript too?
The figure 9 provides a good scheme. Since it is given in the discussion, it would be great to add information, if any of the indicated enzymes are known as direct targets of resveratrol. It could be given as an asterisk with a citation in the legend for example.
Author Response
We thank again the reviewer for his/her constructive comments. In the new version of the manuscript we have addressed all of the comments as follows:
Reviewer: First of all, you didn't check normality of the data. Could you state clearly, if the data could pass a normality test or not? This is of utmost importance, when you use t-test.
Most of the data, including graphs, are based on the comparison of sets of three data points. Without a statement about normality (or outliers) or adding the exact points on each graph, there still exist a possibility, that some statistical results might be concluded from the effect of e.g. outliers.
Answer: The InStat software we used automatically performs a data normality test. All data related to the investigated amounts of sphingolipids and enzyme activities passed this test. The results from the immunoblottings, where we used pooled samples did not provide sufficient number of data points to perform normality test. According to the reviewer’s suggestion we added the exact points on the graphs, presenting the immunoblot results (Figs. 2, 5, 6, 7 and 8). These clarifications are included in the Statistical analysis section as follows: “All data related to the investigated amounts of sphingolipids and enzyme activities passed normality test. The results from the immunoblottings where pooled samples were used, did not provide a sufficient number of data points to perform this test. Therefore, the specific values from each independent experiment are indicated on the corresponding figure.”
Reviewer: You wrote in the answers: "The figures presented in the manuscript were obtained from three independent experiments, e.g. from experiments performed on three different days with a total of nine cell lysates for the control and as many for each treatment. In immunoblotting experiments, lysates from each independent experiment were pooled so that all preparations could be loaded onto a single gel and treated under the same antibody conditions.".
This explanation, containing a statement about 9 cell lysates (and their pooling) is better, than the current description, which contains "three replicate samples" and "data were obtained from three independent experiments performed on different days and containing three replicates for the control and each of the treatments". Could you indicate of the 9 cell lysates in the manuscript too?
Answer: According to the reviewer’s suggestion we have included the following statement in the Statistical analysis section: The analyzed data for each figure were obtained from three independent experiments, e.g. from experiments performed on three different days, each of them containing three replicates, finally resulting in a total of nine cell lysates for the control and as many for each treatment.
In immunoblotting experiments, lysates from each independent experiment were pooled so that all preparations could be loaded onto a single gel and treated under the same antibody conditions.”
Reviewer: The figure 9 provides a good scheme. Since it is given in the discussion, it would be great to add information, if any of the indicated enzymes are known as direct targets of resveratrol. It could be given as an asterisk with a citation in the legend for example.
Answer: Following the reviewer’s suggestion, we updated Fig. 9 and its legend by adding published information, describing changes in the activity of the enzymes from the sphingolipid pathway in response to resveratrol, obtained for different cells.
Reviewer 2 Report
The Authors have added some new references in the introduction of the paper as I had suggested, and have inserted the figure 9. They have corrected some omissions and incorrectness. They have answered my questions comprehensively and have explained that this paper is part of a more complete research whose other results are also reported in another submitted paper or are ongoing. Consequently it is possible to better evaluate the findings reported in the paper submitted to IJMS. In my opinion I can suggest to the Editor that the paper can be considered for the publication.
Author Response
My collaborators and I thank this reviewer for his/her positive opinion and pertinent remarks, which undoubtedly improved the quality of our paper.